# The Impact of Qualification and Hospice Education on Staff Attitudes during Palliative Care in Pediatric Oncology Wards—A National Survey

**DOI:** 10.3390/children11020178

**Published:** 2024-02-01

**Authors:** Eszter Salamon, Éva Fodor, Enikő Földesi, Peter Hauser, Gergely Kriván, Krisztina Csanádi, Miklós Garami, Gabor Kovacs, Monika Csóka, Lilla Györgyi Tiszlavicz, Csongor Kiss, Tímea Dergez, Gábor Ottóffy

**Affiliations:** 1Division of Pediatric Hematology and Oncology, Department of Pediatrics, University of Pécs Medical School, József A. Street 7, 7623 Pécs, Hungary; fodor.eva@pte.hu (É.F.); ottoffy.gabor@pte.hu (G.O.); 2Institute of Behavioural Sciences, Semmelweis University, Nagyvárad tér 4, 1089 Budapest, Hungary; 3Pediatric Center, Semmelweis University, Tűzoltó utca 7–9, 1094 Budapest, Hungary; hauser.peter@dpc.korhaz.hu (P.H.);; 4Velkey László Child’s Health Center, Borsod-Abaúj-Zemplén County Central Hospital and University Teaching Hospital, Szentpéteri kapu 72–76, 3526 Miskolc, Hungary; 5Department for Pediatric Hematology and Hemopoietic Stem Cell Transplantation, Central Hospital of Southern Pest, National Institute of Hematology and Infectious Diseases, 1097 Budapest, Hungary; 6Hemato-Oncology Unit, Heim Pál National Pediatric Institute, 1089 Budapest, Hungary; haema@heimpalkorhaz.hu; 7Department of Pediatrics, University of Szeged, 6720 Szeged, Hungary; 8Department of Pediatrics, Faculty of Medicine, University of Debrecen, 4032 Debrecen, Hungary; kisscs@med.unideb.hu; 9Institute of Bioanalysis, University of Pécs, 7624 Pécs, Hungary; timea.dergez@aok.pte.hu

**Keywords:** palliative care, pediatric oncology, Hungary, hospice care, compassion fatigue, compassion satisfaction, burnout, narrative psychology, medical communication, nationwide

## Abstract

Background: Our knowledge about the attitudes of healthcare staff to palliative care in pediatric oncology is scarce. We aimed to assess their perceptions of palliative care in Hungary and find answers to the question of how to provide good palliative care for children. Method: Physicians (*n* = 30) and nurses (*n* = 43) working in the field of pediatric oncology (12 of them specialized in hospice care) were interviewed. Palliative care practice (communication, integration of palliative care, professionals’ feelings and attitudes, and opportunities for improvement) was assessed by semi-structured interviews evaluated in a mixed quantitative and qualitative way by narrative categorical content analysis and thematic analysis. Results: All providers displayed high negative emotions, positive evaluations, and used many active verbs. Nurses showed higher levels of denial, more self-references, and were more likely to highlight loss. Physicians emphasized the importance of communication regarding adequate or inadequate palliative care. Hospice specialists showed a higher passive verb rate, a lower self-reference, a lower need for psychological support, and a greater emphasis on teamwork and professional aspects. Conclusion: Our results show that nurses are more emotionally stressed than doctors in palliative care in pediatric oncology. To our knowledge, a study comparing doctors and nurses in this field has yet to be carried out. Our results suggest that pediatric oncological staff can positively evaluate a child’s palliative care despite the emotional strain. Regarding hospices, professional practice in palliative care may be a protective factor in reducing emotional distress and achieving professional well-being.

## 1. Introduction

Palliative care is a multidisciplinary approach that aims to improve the quality of life of patients with life-limiting illnesses and their relatives, whether or not the patient receives curative treatment [1,2,3]. Specifically referring to pediatric palliative care (PPC), the WHO says, “palliative care for children is the active, holistic care of the child’s body, mind and spirit, including support for the family” [3]. Although oncology is a rapidly evolving field of medicine, malignancies remain the leading cause of death in children and adolescents [4]. Recent surveys show that although mortality rates vary, children’s palliative care needs are similar across Europe. The foundations for European pediatric palliative care were laid in Trento in 2006. They have evolved rapidly over recent decades, and several standards and guidelines have been published [5,6,7,8,9,10,11]. Key aspects include the following: All children with life-limiting, life-threatening, or terminal illnesses are eligible for PPC. PPC should improve the quality of life and consider the needs, choices, and wishes of children and their families. PPC should not be limited to end-of-life care but should be provided at the time of diagnosis of a life-limiting or life-threatening condition, or in some cases before diagnosis, when it may be challenging, for example, because of advanced illness. The level of care should be determined according to the child’s and family’s specific needs and may vary over time. There are different levels of palliative care (palliative approaches used by all healthcare providers, general PPC, specific PPC), which professionals should provide with the appropriate level of PPC training [12]. In line with global changes, there have also been recent developments in pediatric palliative care in Hungary [13]. However, some gaps need to be highlighted. In Hungary, the concept of PPC is strongly conflated with end-of-life care. The early integration of PPC, the definition of palliative care, the lack of protocols within the country, the development of multidisciplinary palliative teams, and the possibility of home hospice remain significant challenges.

The general purpose of this study was to explore the perspectives of pediatric oncology healthcare providers working in Hungarian pediatric oncology institutions regarding the current state of palliative care delivery for children from the first visit to end-of-life care. More specifically, we aimed to investigate whether the experiences or perceptions differed from the perspective of nurses vs. doctors and hospice-trained vs. non-trained providers through narrative content analysis methods. In Hungary, the concept of PPC is strongly conflated with end-of-life care, and this conflicting terminology was primarily used in the current article.

## 2. Materials and Methods

### 2.1. Sampling Considerations and Recruitment

Subjects were doctors and nurses working in seven pediatric oncology centers in Hungary (I–VII, n = 73). Twelve interviewees worked in hospice care in addition to their departmental work. During the interview process, we aimed to have an equal representation of doctors and nurses from each department. We spent 2–3 days in most wards and interviewed almost all staff members on shift at that time. The recruitment criterion was that the person had worked in the department. There were no exclusion criteria. Respondents involved in palliative care were not explicitly selected (i.e., colleagues who had received palliative/hospice-oriented training in addition to their training in pediatric oncology and were involved in hospice care at the time of the interview). One of the most essential differences in their participation is that they mainly work in the same place, as only one of the seven pediatric oncology centers in the country currently has a hospice team. At the time of the interview, children’s hospice care in Hungary was provided in three places: in a children’s hospice house in the capital, a home care program in the capital, and a mobile hospice team in one of the seven children’s oncology centers. Providers were recruited voluntarily, regardless of age, gender, and years spent in the department. Staff were initially contacted in person, then by phone, and then online because of the 2020 COVID-19 pandemic. The most significant difficulty of the COVID-19 pandemic was that the interview process slowed down considerably. The extra workload on health workers made it more difficult to recruit participants, and less free time made it harder for doctors and nurses to say yes to requests. After selection, the results of the face-to-face and online interviews were similar. There are several reasons for this. The interviewer was always the same person, and, as with the face-to-face interviews, the online interviews were preceded by some introductions and informal discussions to help obtain honest answers. The questions stayed the same. The interview was about a complex topic, which was the same, and the distance did not change it.

### 2.2. Ethical Approval

The research ethics license for the study was issued by the Scientific and Research Ethics Committee of the Health Science Board (SSC TUKEB). Case No 47846-6/2018/EKU.

### 2.3. Data Collection Tools

A semi-structured interviewing technique with 33 questions was used during the study to document staff’s opinions and sentiments toward palliative care (Appendix A, Appendix A). The interviews were audio-recorded on a tape recorder and then transcribed verbatim. The interviews were coded after the audio recordings. The recordings were then processed anonymously, and the full interviews were only made available to the colleagues directly involved in the verbatim transcription, one of whom was the interviewer herself. The shortest interview lasted 11 min, and the longest lasted 55 min. In addition, before each interview, some data were recorded concerning the age of the staff member, the number of years spent in the ward, their position, and their participation in hospice therapy. Interview questions covered the following main topics:(1)Introduction to palliative therapy, first communication;(2)System of palliative care, advantages and disadvantages of palliative therapy at home and in the hospital;(3)The communication of the terminal state;(4)Most important aspects of palliative therapy from the perspective of the child, parents, and caregivers;(5)What makes palliative care appropriate and successful.

### 2.4. Coding and Analysis

The interviews were analyzed anonymously. The interviews were analyzed in two ways: first using narrative categorical content analysis (NarrCat), and then thematic analysis. The results were grouped by qualification (doctor, nurse); gender (male, female); center; age; hospice attendance; and years in the ward.

Narrative categorical analysis and its toolkit, NarrCat, were initially developed by the Hungarian Narrative Psychology Group. NarrCat is an automated content analysis tool that uses a computer to convert narrative sentences into psychologically relevant, statistically processable narrative categories. The content analysis software used in scientific narrative psychology is NooJ, a multilingual language development environment [14]. The international NooJ community comprises 19 languages, including several with non-Latin scripts. NooJ’s usability for analyzing Hungarian texts is based on the Hungarian National Corpus [15]. When a person tells a story, he or she expresses his or her relationship to the world, self, and social relationships [16]. Narrative content analysis aims to reveal the relationship between a narrative’s content and structural elements and the correlation with psychological functioning to make the results quantitatively testable. As with all content analysis research, narrative psychological content analysis begins with qualitative judgments that assign meaning to text elements. This meaning is usually psychological. However, the analysis goes beyond this qualitative phase. Narrative psychological content analysis treats content analysis codes as values of psychological variables that become quantifiable and statistically processable [17,18]. As can be seen from the above, narrative content analysis helps us to infer the psychological factors of the storyteller, regardless of the content of the narrative. We have chosen this analysis method to assess and present this otherwise subjective and challenging topic in a quantifiable way. The current research used six program modules: agency, evaluation, emotional, spatiotemporal perspective, social reference, and negation modules. These were used to infer the extent to which each narrator was an active or passive provider in the care, whether they experienced positive or negative emotions, their social relations, their level of denial, and their ability to process the events they experienced. In the agency psycho-thematic module, the active verb was interpreted as the desire for healing, the will to do something, and the need for workers to experience their actions as effective in therapy. There were also situations where it referred to a desire for increased control over events. The frequency of using the passive verb was explained by a tendency to withdraw into the background, emphasize the patient, and reduce the workers’ role. The emotions module was interpreted more as unconsciously experiencing positive and negative emotions. In contrast, the evaluation module was interpreted as the evaluation following conscious processing. For the spatio-temporal aspect modules, we assessed the processing of the events narrated based on three perspective forms. We assessed the events as processed using the retrospective perspective form, less processed using the experiencing perspective form, and least processed using the metanarrative perspective form. For the social referents, We-reference was used as an indicator of identification with the group and attentiveness to the group. In contrast, Self-reference was associated with a desire for increased control. The negation module was interpreted as negativism, signs of emotional distress, and having a defense mechanism. The reliability of individual modules has been the subject of several studies [19], but a detailed discussion of these is beyond the scope of this paper. Instead, Appendix B discusses the interpretation of the modules used in more detail.

Our second method of analysis was a thematic analysis. Two different people read the interview responses and assigned codes to them. They then reviewed the codes they had created and developed themes based on the patterns between them, which varied according to the nature of the questions (simple choice or open-ended) (e.g., the question is as follows: If you had the opportunity to change three things about how your ward provides palliative care, what would those three things be? The themes are as follows: developing infrastructure (1), which includes more palliative specialists/teams, training, staffing, ward/department, and home hospice; increase in psychosocial support (2), which includes psychosocial support for staff and patients; earlier introduction of palliative therapy (3), which includes all the phrases related to starting earlier, not delaying it, and talking about it; and developing protocols and definitions (4), including what/when/and why we give.). From the 33 questions, we analyzed the answers of the most relevant eight questions.

### 2.5. Statistical Analysis

All statistical analyses were performed using IBM SPSS Statistics 28 Software [20]. A descriptive statistical analysis was performed based on our examined sample. For the NarrCat analysis results, Mann–Whitney U-tests were used to compare the number of words across different groups based on center, position, gender, and existence of hospice. In cases of more than two examined groups, we used independent-sample Kruskal–Wallis tests. The categorical data of the sample were analyzed using contingency tables and chi-squared or Fischer’s exact tests. Spearman’s rank correlation and coefficients were defined to examine the relationship between two variables as necessary, and between first-person singular and positive emotions and first-person plural emotions. Differences and correlations were considered significant at *p* < 0.05.

## 3. Results

### 3.1. Sample Characteristics

The first table shows the summary of the sample (Table 1). A total of forty-three nurses and thirty-two doctors took part in the interviews. For two physicians, the audio recordings of the interviews were unsuccessful for technical reasons, so these interviews were excluded from the study. Our study sample consisted of 73 interview subjects in total. Of these, 59% of the respondents were nurses (n = 43) and 41% were physicians (n = 30). There were no significant differences in the proportions of assignments between centers; 16.5% of the subjects (n = 12) worked in hospice care, while 83.5% (n = 61) did not. Of the sample, 13.3% (n = 4) of doctors and 18.6% (n = 8) of nurses participated in hospice care, with no significant difference between the proportions of the groups. Regarding gender, 87.7% of respondents (n = 64) were female and 12.3% (n = 9) were male. There was no significant difference in the gender proportions across the different centers, with women predominating. Our group was also homogeneous in terms of age and number of years spent in the ward, with no significant differences within or between centers (Appendix A, Appendix A).

### 3.2. Results of NarrCat Analysis

Table 2 shows the summary of our significant results (Table 2). The complete data set is shown in the Appendix A (Appendix A).

### 3.3. Results of the Thematic Analysis

The results were analyzed along the lines of the questions asked. The results of the simple choice questions are presented in Table 3 and Table 4 (education and hospice education). Responses to the open-ended questions are presented in Figure 1, Figure 2, Figure 3, Figure 4 and Figure 5, along with an explanation of the themes that emerged from the responses. Tables showing the results for the total sample are provided in the Appendix A (Appendix A).

*“What are the three most common difficulties in order of frequency in implementing palliative care?”* (Figure 1). Based on the most common responses, the themes are as follows: (1) Lack of infrastructure: This includes lack of wards, department facilities, staffing, medicine, and equipment. (2) Lack of definition of palliation: Includes lack of protocols, when/what/why to give, and when to introduce palliation. (3) Psychological burden: Psychological stress on staff, patients, and relatives. (4) Pain relief.

*“If you had the opportunity to change three things about how your ward provides palliative care, what would those three things be?”* (Figure 2). The themes are as follows: (1) Developing infrastructure: Includes more palliative specialists/teams, training, staffing, wards/departments, and home hospice. (2) Increase in psychosocial support: This includes psychosocial support for staff and patients. (3) Earlier introduction of palliative therapy: Includes all the phrases related to starting earlier, not delaying it, and talking about it. (4) Developing protocols and definitions: Includes what/when/and why we give.

*“When do you consider a child’s palliative care appropriate?”* (Figure 3). The themes are as follows: (1) Support of the child’s needs: Includes not being in pain, not suffering, not being afraid, having people around them whom they want, and being able to talk about it. (2) Preparation of the parents/communication: Includes the parent being accepting, prepared, able to stay calm, able to let go, and able to be there for their child. (3) Timely introduction of palliative therapy: Includes having time to be at home, prepare, say goodbye, and time for family to accept the situation (i.e., it is not sudden). (4) Team/professional aspects: Includes team communication, working, supporting each other, and doing all they can professionally. (5) Undecided.

*“When do you consider a child’s palliative care inappropriate?”* (Figure 4). The themes are as follows: (1) Lack of needs of the child: Includes feeling pain, being scared, suffering, not having loved ones around, and not understanding what was happening. (2) Lack of communication: Includes the parent not being prepared, wanting something else, or not cooperating. (3) The belated introduction of palliative care: Includes being late, having no time to prepare, and sudden death. (4) Team/professional aspects are poorly implemented: Poor team communication, pain relief, and wrong medication. (5) Undecided. (6) There is no such thing as inappropriate.

*“What is the most important issue about the dying or death of a child from your perspective?”* (Figure 5). The themes are as follows: (1) Supporting the parent’s needs, including being able to accept, move on, and go peacefully. (2) Supporting the child’s needs, including being able to die peacefully, quietly, without pain, and being able to do what they wanted. (3) Team/professional aspects, including that staff can move on and that everything is done well. (4) Clear communication, including starting palliative therapy on time, saying things, not hiding things, and effectively debriefing the family. (5) Coping with the fact of death: This includes expressing grief, not asking questions, feeling helpless, and feeling bereaved.

## 4. Discussion

The general purpose of this study was to explore the perspectives of pediatric oncology healthcare providers working in Hungarian pediatric oncology institutions regarding the current state of palliative care delivery for children. More specifically, we aimed to investigate whether the experiences or perceptions differed from the perspective of nurses vs. doctors and hospice-trained vs. non-trained providers. We hypothesized that more skilled hospice providers would have a more positive experience of palliative care.

Several studies highlight the importance of education and report that higher education also promotes coping mechanisms for staff [21,22]. It is widely documented in the literature that a lack of education may be a trigger for maladaptive coping mechanisms at the time of patient death. Education is essential to improve health care and ensure professionals’ physical and psychosocial integrity [23]. Our results are consistent with these studies. Many of our findings on participation in hospice care suggest that those closer to palliation can more easily shift the focus to the patient so that self-defense mechanisms can better function and boundaries can be maintained more easily. In addition, the findings on social aspects and the emphasis on team and professional aspects suggest a greater sense of belonging to a group, retaining the individual’s power. The higher education of staff also allows them to focus on the professional aspects of delivering care, because if we know what needs to be done, we know what makes care feel right. Our results show that this knowledge can also help staff perceive palliative care delivery as successful. To our knowledge, limited studies have compared providers with different hospice training working in an oncology department. This subject could be an essential research topic in the future.

The literature on the stress experienced by pediatric oncology staff is limited [23]. Nevertheless, researchers in recent decades have recognized that the death of a child can be traumatic not only for the family, but also for the healthcare professionals responsible for the child. However, a literature review highlights that palliative care professionals report that their work is meaningful and that caring for patients at the end of life can be a rewarding experience. There is also evidence to suggest that involvement in palliative care can help professionals cope with death and loss [24,25,26,27,28,29,30]. The results of the evaluation and emotion modules, narrative perspectives, and the “success/failure” question may suggest that workers can positively evaluate palliative care. One interviewee said, “I cannot tell you a specific difficulty because what I do is not a difficulty. For me, it is a huge blessing to accompany them to the end, to say goodbye to them, and to do everything I can to ensure they leave without pain and have a good time.” This result aligns with the literature data showing that palliative care workers often report high satisfaction and find their work rewarding. [31,32] Our essential finding is that although palliation and death experience result in loss and negative emotions, there is room for a positive evaluation. In addition, our significant findings about hospice providers are also consistent with the literature cited above. Pediatric palliative care providers, who are more skilled in hospice care, may experience higher levels of success and a lower need for psychological support, and involvement in hospice care may promote better coping mechanisms and improved team focus in the case of end-of-life care.

On the comparison of doctors and nurses, an important new data point about psychological strain is the difference between workers by job title. We hypothesized that nurses may experience a higher amount of stress. A growing body of literature demonstrates that workers are exposed to the emotional strain of providing palliative care [24,29,30], and our findings are consistent with this. [18,33] The results of our research suggest that nurses are exposed to higher emotional strain than doctors and have more difficulty processing events. Based on a literature review, several studies have shown that pediatric oncology nurses are exposed to high stress. Some studies point to differences between nurses and doctors [34]. To our knowledge, studies directly comparing the pediatric palliative care experiences of doctors and nurses working in pediatric oncology departments are limited, and could be an essential focus in the future.

Some of our secondary findings highlight the communication gap between doctors and nurses. Our results suggest that doctors highlighted parents’ needs and communication with parents more than nurses did, possibly because palliative care usually occurs in the ward, a medically mediated environment. In addition, our findings showed a significant difference between doctors and nurses discussing terminal illness symptoms with the family. One interviewee said, “Well, doctors talk to parents about it.” The literature suggests several barriers to effective teamwork, including the traditional medical model, hierarchy, and lack of collaborative training. A collaborative communication model facilitates the active involvement of patients and families in care, which has been described as one of the foundations of pediatric palliative care [35]. Collaborative teamwork is associated with positive outcomes, including improved communication, improved patient compliance, and, as a result, greater family satisfaction, depending on family-centered care [35]. Based on the literature, building teamwork can also increase the rate of compassion satisfaction [36], so information sharing could be essential to reduce the burden on caregivers. In addition, end-of-life discussions in palliative care have many benefits. The AAP (American Academy of Pediatrics) recommends that patient and family partnership is an integral part of health care, and that adequate information is essential [37,38,39]. It also reduces the stress experienced by parents and allows carers to share the stress they are under [40,41]. Our findings point to a need for more communication between doctor and nurse teams in several areas, which may be one reason why nurses are subject to more significant emotional strain.

One aim of this study was to explore the perspectives of pediatric oncology healthcare providers working in Hungarian pediatric oncology institutions regarding palliative care delivery for children. In Hungary, the concept of PPC is strongly conflated with end-of-life care, and this was the primary focus in the current article. The literature recommends an early integration of palliative care [4,5,6], but this is often difficult, and previous studies have described several barriers to the early integration of palliative care [42,43,44]. One barrier may be the lack of an appropriate definition of palliation, as reflected in our results. Our results suggest that the lack of definition is higher up the list of difficulties identified by staff than the lack of infrastructure (of the results for the whole sample), which illustrates that staff also feel the lack of a definition confuses the concept of PPC with end-of-life care. Furthermore, if palliation is not adequately defined or education about palliative care is low, it can lead to uncertainty and maladaptive coping [45]. During the interviews, only 1 of 73 respondents said that palliative care was present early. In another study in Hungary, the majority of parents who had lost a child said that palliative care was introduced after all possible treatments. Thus, there was a substantial gap between curative and palliative therapy [46]. This study and our present research findings highlight the discrepancy between international recommendations and current practice and draw attention to the need to rethink the concept of palliative care.

## 5. Limitations

One limitation of the study is that palliative care needs to be more well defined in the country. Even among professionals, the concepts of hospice, end-of-life care, and palliative care need to be clarified. During the study, we ourselves accepted the confusion of the latter two concepts in our country. The interviews do not directly refer to the participants’ understanding of PPC, but we did not intend to create a new or exact definition. This article intended to contribute to the discussion of palliative care by analyzing the country’s current state. The doctor-to-nurse ratio was unequal among the participating centers. In addition, the interviews were recorded over about three years. During that time, we had to switch from face-to-face to online interviewing due to the COVID-19 pandemic, which may also have affected the results. To avoid differences, the interviewer did not change from the beginning of the study. A limitation of the research is that no validated questionnaire was used for the evaluation. The interview questions were constructed based on subjective interest, and narrative content analysis was used to answer the psychologically assessable characteristics of the narrator. Finally, although narrative content analysis can theoretically evaluate any text, it is most effective for longer, single-context storytelling. To avoid bias, we sought to keep the interviewers talking for as long as possible, and we also paid attention to the number of words used in the analysis.

## 6. Conclusions

To our knowledge, no previous study has used narrative content analysis to assess the well-being of pediatric oncology workers. In contrast to a self-assessment questionnaire, narrative analysis examines providers subliminally. In our results, the values of narrative analysis and thematic analysis supported each other. Based on these results, further application of the method could help assess professionals.

It is important to emphasize that the results show that nurses are more emotionally stressed than doctors. There are several possible reasons why nurses are under more stress, such as spending more time directly with the patient, caring for the dying person with the parent, or being in more situations where they need to communicate with the parent. In a hierarchical system, they have less influence to shift the family increasingly from curative to palliative therapy, which can lead to a feeling of helplessness. To our knowledge, a study comparing doctors and nurses concerning pediatric palliative care has not yet been carried out, and this could be an essential topic for future research.

Every practitioner working in a pediatric oncology department encounters children’s deaths. Although palliative care is available in all pediatric oncology departments, the number of professionals who assist in palliation is limited. Our results suggest that professionals with higher palliative training are more likely to experience palliative care as a success among those working in pediatric oncology. Our study suggests that engagement with hospice care positively impacts communication and focus on professional and team aspects. Palliative education may reduce the need for psychological support and reduce staff uncertainty. To our knowledge, limited studies have compared providers with different hospice training working in an oncology department. This subject could be an essential research topic in the future.

Our study examined several factors related to pediatric oncology care in Hungary. Based on the interviews with the staff, we found that the Hungarian pediatric palliative care system is not uniform. The early integration of palliation remains difficult. Nurses are exposed to high psychological stress, which is further complicated by the lack of a clear definition of palliative care and little education on palliation. However, our results suggest that staff can positively assess a child’s palliative care despite the emotional strain. We have shown that teamwork contributes to positive emotions. Our findings conclude an urgent need to improve palliative care among staff. It is essential to establish national guidelines to better define palliative care. We encourage doctors and nurses to participate in hospice-related training to expand their knowledge and experience. This includes, for example, a palliative licensing exam for doctors, or hospice nurse training for nurses. We recommend increasing palliative training for staff and setting up palliative care teams, as this would positively impact patients, their families, and healthcare providers.

## Figures and Tables

**Figure 1 children-11-00178-f001:**
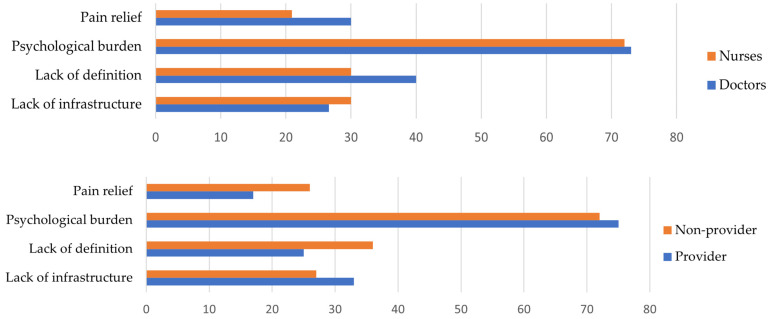
*“What are the three most common difficulties in order of frequency in implementing palliative care?”* (by hospice education and qualification). There is no significant difference, but notably, psychological distress is the most frequently mentioned difficulty in all groups.

**Figure 2 children-11-00178-f002:**
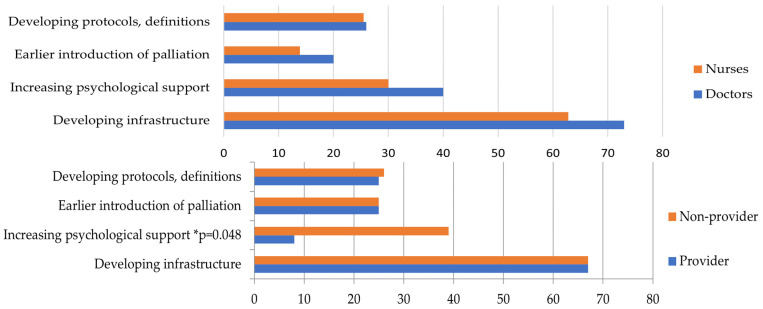
*“If you had the opportunity to change three things about how your ward provides palliative care, what would those three things be?”* (by qualification and hospice education). There is no significant difference by job title, and the order is the same. In both groups, “developing infrastructure” was the most frequently mentioned change. In hospice comparison, “increasing psychological support” showed significance, with a lower value for hospice providers than non-providers. * Significant results.

**Figure 3 children-11-00178-f003:**
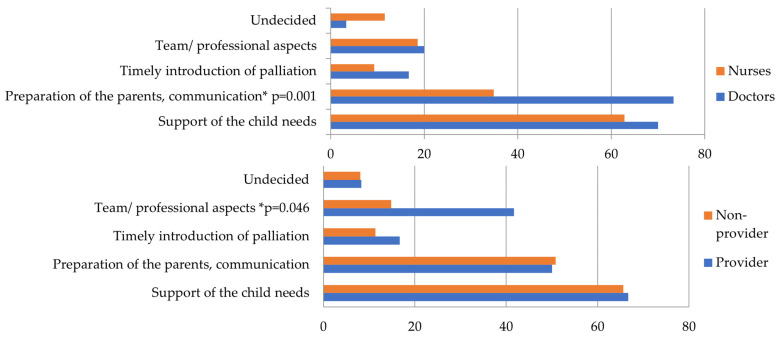
*“When do you consider a child’s palliative care appropriate?”* (by qualification and hospice education). Doctors mentioned “preparation of the parents, communication” significantly more than nurses (*p* = 0.001). On a hospice basis, hospice providers were significantly more likely to mention “Team/professional aspects” than non-providers (*p* = 0.046). * Significant results.

**Figure 4 children-11-00178-f004:**
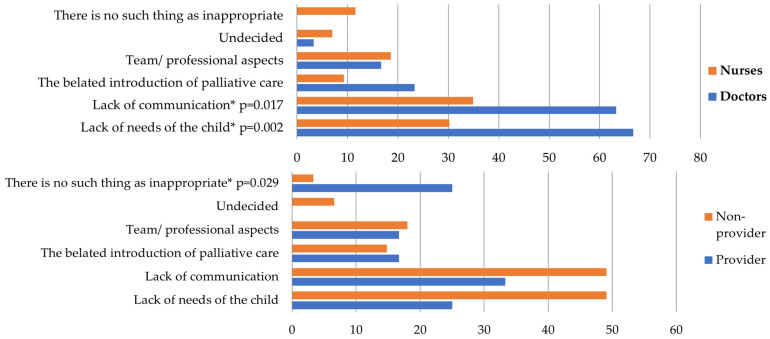
*“When do you consider a child’s palliative care inappropriate?”* (by qualification and hospice education). Doctors also mentioned “lack of needs of the child” (*p* = 0.002) and “lack of communication” (*p* = 0.017) significantly more than nurses. Hospice providers were significantly more likely to say “there was no inappropriate palliation” than non-providers (*p* = 0.029). * Significant results.

**Figure 5 children-11-00178-f005:**
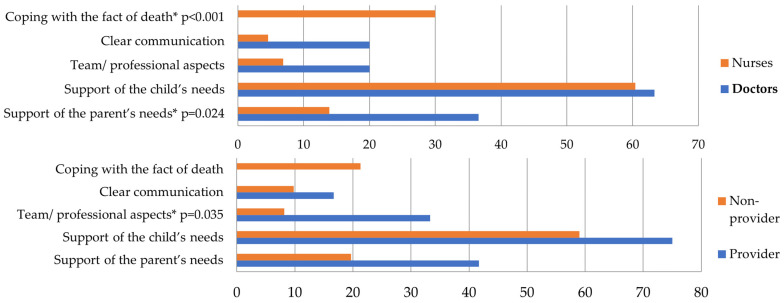
*“What is the most important issue about the dying or death of a child from your perspective?”* (by qualification and hospice education). Based on the qualification, “support of the child’s needs” was mentioned equally by staff, while “support of the parent’s needs” was mentioned more often by doctors than by nurses (*p* = 0.024). The theme of “coping with the fact of death” was represented in 30% of nurses and did not appear for doctors (*p* < 0.001). Example: One interviewee said, “The doctor can leave the ward, if necessary, but we must stand by the bedside.” The nurse group’s commonly used aspect was “coping with the fact of death,” which suggests that the focus is on experiencing pain and loss. One interviewee said “What is the most important question for me? Why do I have to watch this? Why do I have to go through this?” No one in the doctors’ group gave such an answer. Hospice providers mentioned “team/professional aspects” more often than non-providers (*p* = 0.035). As for “coping with the fact of death,” it is notable that none of the hospice providers mentioned this, while 21% of the non-providers did. * Significant results.

**Table 1 children-11-00178-t001:** Sample characteristics.

Total Interviews (73)				
Groups	Sub-Groups	Total Numbers	Subcategories	Percentages
Qualification	Doctors	30	4 hospice providers—26 non-providers	41% (13.3–86.7%)
	Nurses	43	8 hospice providers—35 non-providers	59% (18.6–81.4%)
Hospice	Providers	12	4 doctors—8 nurses	16.5%
	Non-providers	61	26 doctors—35 nurses	83.5%
Gender	Male	9	8 doctors—1 nurse	12.3% (88.2–11.1%)
	Female	64	22 doctors—42 nurses	87.7% (34.4–65.6%)

**Table 2 children-11-00178-t002:** Significant results of narrative analysis.

Grouping	Categories—Median (IQR)	*p* Value
Whole cohort	Active verbs 2.32 (0.94) > passive verbs 0.53 (0.29)	*p* < 0.001
	Constraint 0.58 (0.32) > intention 0.37 80.27)	*p* < 0.001
	Total emotion 1.04 (0.37) > total evaluation 0.93 (80.35)	*p* = 0.025
	Evaluation: positive 0.69 (0.31) > negative 0.21 (0.14)	*p* < 0.003
	Emotion: negative 0.58 (0.32) > positive 0.45 (0.26)	*p* = 0.005
	Positive evaluation 0.69 (0.31) > positive emotion 0.45 (0.26)	*p* < 0.001
	Negative emotion 0.58 (0.32) > negative evaluation 0.21 (0.14)	*p* < 0.001
	Experiential form 16.17 (2.82) > metanarrative and retrospective form 3.43 (2.04)	*p* < 0.001
Dependent on qualification	Self-reference: nurses 3.32 (2.07)> doctors 2.89 (1.76)	*p* = 0.019
	Negation: nurses 4.44 (1.63) > doctors 3.69 (1.29)	*p* < 0.001
	Metanarrative form: nurses 3.64 (2.42) > doctors 3.14 (1.71)	*p* = 0.041
Dependent on hospice education	Passive verb: providers 0.68 (0.56) > non-providers 0.50 (0.27)	*p* = 0.022
	Self-reference: providers 2.05 (3.01) < non-providers 3.23 (1.67)	*p* = 0.033
Correlations	Constraint and Self-reference, Weak negative correlation R = −0.247	*p* = 0.035
	Positive emotion and We-reference, Weak positive correlation R = 0.255	*p* = 0.029

From the numbers obtained by Narrkat, we created ratios concerning the total number of words. These ratios were compared via Mann–Whitney U-tests. The results of the comparisons are given as median (IQR) values.

**Table 3 children-11-00178-t003:** Results of the thematic analysis (by qualification).

Question	Responses	Doctors	Nurses	*p*-Value
*“A question about your experience of losing a child, or more specifically, the journey leading up to it: Have there been times when it was a success, have there been times when it was a failure?”*	It can be a success	73.3%	58.1%	
	It cannot be a success	16.7%	16.3%	*p* = 0.238
	Undecided	10%	25.6%	
*“Are the symptoms of the terminal condition discussed with the family?”*	**Yes**	**93.3%**	**58.1%**	***p* = 0.003 ***
	No	6.6%	25.6%	
	Undecided	0	16.3%	
*“Could the discussion about the terminal state be late with adverse consequences for the patient, the environment or care?”*	Yes	86.6%	86%	*p* = 1.0
	No	6.7%	7%	
	Undecided	6.7%	7%	

* Significant result.

**Table 4 children-11-00178-t004:** Results of the thematic analysis (by hospice education).

Question	Responses	Provider	Non-Provider	*p*-Value
*“A question about your experience of losing a child, or more specifically, the journey leading up to it: Have there been times when it was a success, have there been times when it was a failure?”*	It can be a success	92%	59%	
	It cannot be a success	8%	18%	*p* = 0.081
	Undecided	0	23%	
*“Are the symptoms of the terminal condition discussed with the family?”*	Yes	91.7%	68.9%	*p* = 0.379
	No	8.3%	19.6%	
	Undecided	0	11.5%	
*“Could the discussion about the terminal state be late with adverse consequences for the patient, the environment or care?”*	Yes	75%	88.5%	*p* = 0.207
	No	8.3%	6.6%	
	Undecided	16.7%	4.9%	

## Data Availability

The (derived) data in the study are presented in the article and the Appendix A. Additional data (interview material) are not publicly available due to ethical restrictions but are available upon request (anonymous interviews but personal responses). Data presented in this study are available from the corresponding author upon request.

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
