# Peer review of "The Impact of Qualification and Hospice Education on Staff Attitudes during Palliative Care in Pediatric Oncology Wards—A National Survey"

_children, 2024, doi:10.3390/children11020178_

Round 1

Reviewer 1 Report

Comments and Suggestions for Authors

Dear authors, thank you for allowing me to revise your manuscript focused on describing the attitudes of nurses and doctors towards palliative care in pediatric oncology wards.

Although I found the topic is of interest and the manuscript is well-written, I have recognised some serious flaws to your study that I summarise below.

INTRODUCTION

The first issue that recurs throughout all the manuscript is the lack of a clear understanding of your definition of palliative care. Your sentences and reflections jump over palliative care and death/end of life (page 2, lines 48–57, and also in discussion); but you should refer to the definition you are referring to.

The other major issue is the methodological orientation of your study, which I found very confusing. You have collected qualitative data that were analysed quantitatively to assess (and qualitative research is not assessing but is describing or exploring the lived experiences) the palliative care provided. Moreover, you tried to find differences among groups.

I presume you were a little confused when you designed this study, as for assessing this kind of inferences a quantitative design would be more appropriate. Maybe this confusion led you to use a "semi-structured" interview of 33 questions? And what about the length of the interviews? The shorter lasted 11 minutes, but for answering such an amount of questions I would presume the interviewee just said yes or not (and effectively, in results you have provided just the dicothomic data). Perhaps the use of a questionnaire rather than a "semi-structured" interview would be more appropriate.

Always referring to the method you have used and the results you have presented, I have some concerns about the analysis you have performed. In a qualitative study, you should present the verbatim of participants narrations that give credibility to your study (see Lincoln and Guba '85 trustworthiness criteria), and in this manuscript what I found is a series of inferences coming from a sovra-structured classification (positive emotions/metanarrative) that led the reader to more confusion than answers. Moreover, this results are often commented in the section, and they should just be presented, leaving all the possible assumptions to the discussion section (e.g., The  predominance  of  positive  evaluations  may  suggest  that; Higher  levels  of  negation  by  nurses  may  indicate  a  defense  mechanism; The  lower  self-references  of  hospice  providers  may  indicate  that). And in the same way, in the Content analysis what I found is just a dichotomic yes/no categories that gave me nothing qualitative or more substantiated by participants' own words.

To conclude, even if the purpose of your study was interesting and sustained by a good rationale, the data analysis and presentation of your results were not informative and gave nothing more than a plain cross-sectional study (that is not a bad design, but is labelled as what is it). 

I hope this suggestions help you to improve this and your future researches

Regards

Author Response

Thank you for taking the time to review the manuscript. You will find the detailed answers in the uploaded file and the appropriate corrections in the resubmitted manuscript.

Reviewer 2 Report

Comments and Suggestions for Authors

This study presents a questionnaire-based investigation of pediatric palliative care professionals. It reveals significant insights, including differences in awareness across various professions, underscoring its suitability for publication in this journal.

Was their any difference about the data between face to face and online interview?

Author Response

(The authors gave the same response as above.)

Reviewer 3 Report

Comments and Suggestions for Authors

The manuscript explored the impact of qualification and hospice education on staff attitudes during palliative care in pediatric oncology wards – a national survey. My overall evaluation of the manuscript is positive. There are a number of minor revisions, formal and scientific aspects that should be addressed.

The article is well organized in terms of writing. The title of the article is very interesting. But it is necessary to evaluate the working hours of the personnel involved in the oncology department and its effects on their behavior. It is necessary to add explanations about training courses for empowering doctors and nurses to the text of the article. About anxiety control methods that are defined in the oncology department to manage the condition, some information should also be added to the article. If there are guidelines in this regard by the Ministry of Health or World Health, they should be added to the article in the discussion section. Regarding the variety of drugs that are used in palliative medicine and their effects in controlling conditions.

Author Response

(The authors gave the same response as above.)

Reviewer 4 Report

Comments and Suggestions for Authors

First of all, thank you for allowing me to review your manuscript. This Hungarian study investigates the perceptions of paediatric oncology healthcare staff, with an emphasis on the evaluation of palliative care practices, including communication, integration and feelings of professionals. The main aim is to understand how to improve palliative care for children, shedding light on the emotional stress experienced by nurses and doctors and the possible protective role of palliative care practices.

After reading in depth the manuscript, I would like to make some comments and ask the authors several questions about.

Introduction

- Theoretical/conceptual framework: You might consider including a brief theoretical framework that summarises key theories or concepts related to palliative care and practitioners' experience. This would help to contextualise the study within the wider context of the existing literature.

- While addressing the importance of palliative care and the need to understand practitioners' experiences, it might be useful to include a more explicit rationale as to why this study is crucial. For example, what are the specific gaps in knowledge that this study seeks to address, and how might it contribute to improving palliative care in paediatric oncology?

Material and methods

- I would see fit to incorporate more detail on how palliative care respondents were specifically selected and whether there are any distinguishing features of their involvement.

Please, adds information on how diversity and representation in the sample was ensured, and whether there were specific inclusion or exclusion criteria.

- About the impact of the COVID-19 pandemic: Provides additional details on how the pandemic affected the recruitment and interviewing process, and how potential challenges were addressed.

- Information about the recording of interviews on a tape recorder and their subsequent transcription is useful. You can add information on how the confidentiality of participants was ensured and how transcripts were handled.

- The relevance of content analysis through tools such as NooJ is highlighted. You could add a brief explanation of why these tools are appropriate and how they contribute to the understanding of the results.

Results

- Dependent on qualification in table 2. please correct so that the word ends correctly and the syllable is not broken.

Discussion

- it is noted that nurses experience more emotional stress than physicians, you might explore the possible reasons behind this discrepancy.

- After discussing the difficulties, consider offering suggestions or potential solutions. How could the identified problems be addressed? Are there specific interventions that could improve early integration or reduce psychological stress?

- In the limitations section, the fact that a validated questionnaire was not used may affect the validity and reliability of the measurements.

Reflect, pleas, on how this could have influenced the interpretation of the results and discuss possible limitations in the objectivity and consistency of the responses.

Author Response

(The authors gave the same response as above.)

Round 2

Reviewer 1 Report

Comments and Suggestions for Authors

Dear authors, thank you for allowing me to review this second version of your manuscript describing the attitudes of nurses and doctors towards palliative care in pediatric oncology wards.

Although some improvements are present in this amended version, I continue to find some lack of clarity and consistency between data collection, analysis and presentation throughout the text.

Moreover, a clear attempt to reply to my comments was made exclusively for those regarding the presentation of results. 

Particularly, there is lack of clarity about the concept under study: is palliative care provision or is the death of the children the focus of your study? This continues to be reflected in the data collection instrument (interview guide which continues to jump between palliative care and death/loss) that is long and thus not specific to the content, but just to provide a superficial description (and not an assessment as the authors continue to state in their aim) of the practices implemented and the emotions perceived by HCPs. 

In addition, the method is now more deeply described, but it seems more a data mining quantitative approach rather than a qualitative content analysis. And, even if "Appendix A discusses the interpretation of the modules used more in detail", it provides a series of quantitative analyses rather than discussing how modules were interpreted.

Finally, the recruitment of this sample was convenience sampling, and not a random sampling as presented in the sampling considerations section.

Author Response

Dear Reviewer,

Thank you for your comments and suggestions. For a detailed response, please see the attachment. 

Sincerely
Eszter Salamon
